# Hierarchical analysis of RNA secondary structures with pseudoknots based on sections

**Ryota Masuki[1,2], Donn Liew[2], Ee Hou Yong[2]\***

**1** Department of Physics, The University of Tokyo, Tokyo, Japan, **2** Division of Physics and Applied Physics, School of Physical and Mathematical Sciences, Nanyang Technological University, Singapore, Singapore

\* eehou@ntu.edu.sg

## Abstract

Predicting RNA structures containing pseudoknots remains computationally challenging due to high processing costs and complexity. While standard methods for pseudoknot prediction require $O(N^6)$ time complexity, we present a hierarchical approach that significantly reduces computational cost while maintaining prediction accuracy. Our method analyzes RNA structures by dividing them into contiguous regions of unpaired bases ("sections") derived from known secondary structures. We examine pseudoknot interactions between sections using a nearest-neighbor energy model with dynamic programming. Our algorithm scales as $O(n^2 \ell^4)$, offering substantial computational advantages over existing global prediction methods. Analysis of 726 transfer messenger RNA and 454 Ribonuclease P RNA sequences reveals that biologically relevant pseudoknots are highly concentrated among section pairs with large minimum free energy (MFE) gain. Over 90% of connected section pairs appear within just the top 3% of section pairs ranked by MFE gain. For 2-clusters, our method achieves high prediction accuracy with sensitivity exceeding 0.9 and positive predictive value above 0.8. For 3-clusters, we discovered asymmetric behavior where "former" section pairs (formed early in the sequence) are predicted accurately, while "latter" section pairs do not follow local energy predictions. This asymmetry suggests that complex pseudoknot formation follows sequential co-transcriptional folding rather than global energy minimization, providing insights into RNA folding dynamics.

## Author summary

RNA molecules fold into structures to perform biological functions. However, predicting complex RNA structures known as "pseudoknots" is computationally expensive. Current methods often attempt to calculate the entire structure simultaneously, which requires significant computational resources. In this paper, we introduce a hierarchical approach that simplifies pseudoknot prediction. We break the RNA sequence into smaller "sections" of unpaired bases and calculate the

**Data availability statement:** All source code used to perform the analyses in this study, along with the complete datasets, is publicly available on Github. A permanent, citable version of the code and datasets used for this publication is archived on Zenodo.

**Funding:** R.M. acknowledges support by the GRI programme from Nanyang Technological University and the SVAP program from The University of Tokyo. D.L. and E.H.Y. acknowledges support from Nanyang Technological University, Singapore, under its Start Up Grant Scheme (04INS000175C230), Singapore Ministry of Education through the Academic Research Fund Tier 1 (RG140/22) and Academic Research Fund Tier 2 (MOE-T2EP50223-0014). The funders had no role in study design, data collection and analysis, decision to publish, or preparation of the manuscript.

**Competing interests:** The authors have declared that no competing interests exist.

energy required for these sections to bind locally, rather than solving for the global structure. Our analysis shows that strong local interactions are favored by biology; with over 90% of pseudoknots occurring within the top 3% of the most energetically favorable section pairs. This finding allows us to focus computational effort on the small subset of interactions that are most likely to form pseudoknots, rather than testing every possible combination. Our method achieves $> 90\%$ sensitivity for simple 2-section pseudoknots. However, for complex 3-section pseudoknots, only early-forming connections are predictable. This reveals that RNA does not simply fold into the most stable structure. Instead, folding is sequential, with earlier regions establishing interactions that constrain the final structure before synthesis of the later regions.

## Introduction

Ribonucleic acid (RNA) plays fundamental roles across all life forms, carrying genetic information, regulating gene expression, and performing various catalytic functions. Non-coding RNA, which does not translate into proteins, plays important roles in many catalytic and regulatory cellular processes [1–4]. The biological functions of non-coding RNA have strong connections with its molecular structure, particularly pseudoknots [5–8]. Thus, predicting and understanding the structure of RNA for any given base sequence is important in biology, medicine, pharmacy, and other related fields. A fundamental challenge in RNA structure prediction is defining which interactions constitute the secondary structure versus pseudoknots. RNA secondary structure has multiple definitions in literature [9–11]. In this work, we define RNA secondary structure as nested base pairs that can be represented without crossing lines, while pseudoknots are non-planar interactions that create crossing patterns [12–16]. While no universal criterion exists for distinguishing secondary structure from pseudoknot base pairs [17,18], our definition enables computational tractability, aligns with standard structural notations (e.g. dot-bracket format), and provides a framework for our hiearchical method of identifying unpaired sections and predicting pseudoknot interactions between them. RNA secondary structure forms the basic frame of an RNA structure, typically exhibiting greater thermodynamic stability than pseudoknot interactions. Several reliable algorithms have been developed to predict RNA secondary structure for any given base sequence, such as dynamic programming algorithm (DPA) [19–24] and stochastic methods [25–27]. DPA is one of the most widely used methods and has been applied in many different areas [28,29]. DPA accurately predicts RNA secondary structures by minimizing the free energy contribution under a local energy model in $O(L^3)$ time for an RNA sequence of length $L$.

Predicting RNA structures that include pseudoknots remains challenging due to computational complexity and high processing costs. The most straightforward approach would be to DPA to find minimum free energy structures across all possible base pairings. However, this approach is NP-complete even with simple energy models that only consider individual base pair contributions [30–34]. In response,

algorithms have been developed that focuses on specific subsets of pseudoknotted structures [35–40]. These methods improve efficiency by eliminating physically impossible configurations early in the analysis. For example, a dynamic programming algorithm achieving $O(N^6)$ time complexity and $O(N^4)$ space complexity was introduced [10], providing the first method to fold optimal pseudoknotted RNAs using standard thermodynamic models. Though widely used, this approach becomes computationally intensive for longer RNA sequences. A later study showed that simpler dynamic programming techniques could achieve $O(N^5)$ time complexity [41], offering improved performance but lacking the ability to handle complex pseudoknot configurations like kissing-hairpins, which are commonly found in RNA structures. Alternative approaches have emerged, including stochastic methods [42] and novel computational strategies [43,44]. While these techniques show promise in specific scenarios, comparative analyses [45] reveal accuracy limitations, particularly when dealing with complex pseudoknot structures [46].

In order to overcome the trade-off between the accuracy of the model and the computational cost, there is now an increasing focus on hierarchical folding method [40,47,48]. The hierarchical folding method uses a two-step approach: first predicting the pseudoknot-free secondary structure, then calculating pseudoknot base pairs based on this foundation. A proposed DPA variant [49] finds pseudoknotted structures in $O(N^3)$ time and $O(N^2)$ space by applying this hierarchical principle, matching the time efficiency of algorithms that only predict secondary structures. This approach can handle complex configurations like kissing hairpins, which traditionally required $O(N^6)$ time to process. However, comparative studies show these methods face significant challenges with accuracy, particularly for complex pseudoknots in longer sequences [36,45,50]. These limitations appear largely due to the fact that pseudoknot structures are often long-range and thus affected by the many possible 3D RNA conformations, which are difficult to account for in current models.

In this paper, we analyze local pseudoknot interactions by examining how different "sections" of an RNA (derived from known secondary structures) are connected via pseudoknots. Using a nearest-neighbor energy model and dynamic programming, we calculate the minimum free energy of potential pseudoknot formations between unpaired regions. Our approach focuses strictly on local energy contributions between sections and excludes interactions between different pseudoknots.

Building on this section-wise interaction framework, we develop a computationally efficient method that scales as $O(n^2 \ell^4)$, where $n$ represents the number of sections and $\ell$ represents their typical length. This hierarchical approach provides further understanding into how local structural elements contribute to global RNA configurations.

## Methods

### Definitions of terms

We define the terminologies used in the research of RNA structure.

1. **Primary structure** of an RNA refer to its chemical sequence.
2. **Secondary structure** of an RNA is the set of all local nested pairing of complementary bases that can be represented in a planar graph without crossing lines. Even pairings between distant bases (such as initial and final bases) are considered part of the secondary structure if they maintain this planar property. Common secondary structure motif includes hairpin loops, hairpin stems, helical duplexes, bulges, internal loops, and multi-loops. Internal loops consist of unpaired nucleotides connecting exactly two helical segments ($h = 2$), whereas multibranch loops are junction points where three or more helices converge ($h \geq 3$).
3. **Pseudoknots** are base pairings between secondary motifs that cannot be represented without crossing lines in a planar path. These crossing interactions are tetiary structures. Pseudoknots are defined by these crossing patterns within the global RNA topology, not as isolated elements. The non-planarity emerges from the relationship between these interactions and the secondary structure, reflecting the topological complexity characteristic of pseudoknots.

4. **Pseudoknot base pair** is a base pair that participates in a non-nested pairing arrangement, creating crossing inter-actions when represented in a planar diagram. These pairs are part of the global topological structure where bases *i,j* form a pair and bases *k,l* form another pair such that $i < k < j < l$, resulting in crossing lines in a 2D representation.

5. **Pseudoknotted secondary structure** is defined as the set of all base pairs, including pseudoknots. It is specifically used when the set of base pairs includes pseudoknots.

The definitions above are explained graphically in Fig 1A. Next, we will introduce terminologies to be used in our analysis.

1. A **section** is a contiguous subset of bases in the RNA sequence that are not involved in base pairing, as determined from the known structures in our dataset. Specifically, if the *i*th base and *j*th base ($i < j$) are not paired in the sec-ondary structure, they belong to the same section if all bases between them (from position $i + 1$ to $j − 1$) are also not involved in base pairing. Each section represents a single-stranded region that may potentially form pseudoknot interactions with other sections. This definition relies on having accurate secondary structure information, which in our study comes from established RNA structure databases.

2. **Section pair** refers to a pair of sections. The *i*th section and *j*th section are **connected** if there exists a pseudoknot that connects a base in *i*th section and another in *j*th section. See Fig 1B for a graphical description of sections.

3. ***N*-cluster** is a cluster consisting of *N* sections connected by pseudoknots. The number of topologically distinct pseudoknot structures increases rapidly once $N \geq 3$. We will only consider 2-clusters and 3-clusters in this work. The different types of 3-clusters are shown in Fig 1C.

4. **Pseudoknot order** is a metric for RNA structural complexity defined by the minimum number of base pair decompo-sitions to create a nested structure [9]. In our work, 2-clusters correspond to pseudoknot order 1 (1 decomposition; e.g. H/K-type) while 3 clusters correspond to pseudoknot order 2 (2 decompositions; L/M-type).

## Energy model

We use the nearest-neighbor energy model which is based on the chemical and thermodynamic properties of the RNA chains. Nearest-neighbor energy model is widely used to predict secondary structure in conjunction with DPA [51–58]. In the nearest-neighbor energy model, values of free energy contribution are assigned to each closed loop (e.g. stackings, bulge loops, interior loops, multi-loops) in RNA secondary structure, and the total free energy of the structure is calculated as the sum of all the free energy contribution from the loops [59]. The parameters and detailed rules in the energy model used in our analysis are based on mfold version 3.6 [20,21]. While these parameters were derived from experiments in salt-free conditions, we apply them as an approximation for pseudoknot prediction in the absence of comprehensive energy tables calibrated for varying salt concentrations.

Formally, the energy model is defined as

$$E(i,j) = \sum E_{\text{stack}}(s) + \sum E_{\text{bulge}}(b) + \sum E_{\text{interior}}(n) + \sum E_{\text{multi}}(m) + \sum E_{\text{hairpin}}(h) \quad (1)$$

Using Eq 1, we calculate the free energy contribution of the pseudoknot between two sections in the example shown (in linear representation) in Fig 2. According to the conventional rule of the nearest-neighbor energy model, the free energy contribution of the pseudoknot in Fig 2A is given by:

$$
\begin{aligned}
&\text{(FE contribution of the pseudoknot in Fig 2A)}\\
&= -(\text{hairpin loop (1)}) - (\text{hairpin loop (2)}) + (\text{multi loop (3)}) + (\text{stacking (4)}) + (\text{bulge loop (5)}) \quad (2)\\
&+ (\text{stacking (6)}) + (\text{multi loop (7)}) + (\text{stacking (8)}).
\end{aligned}
$$

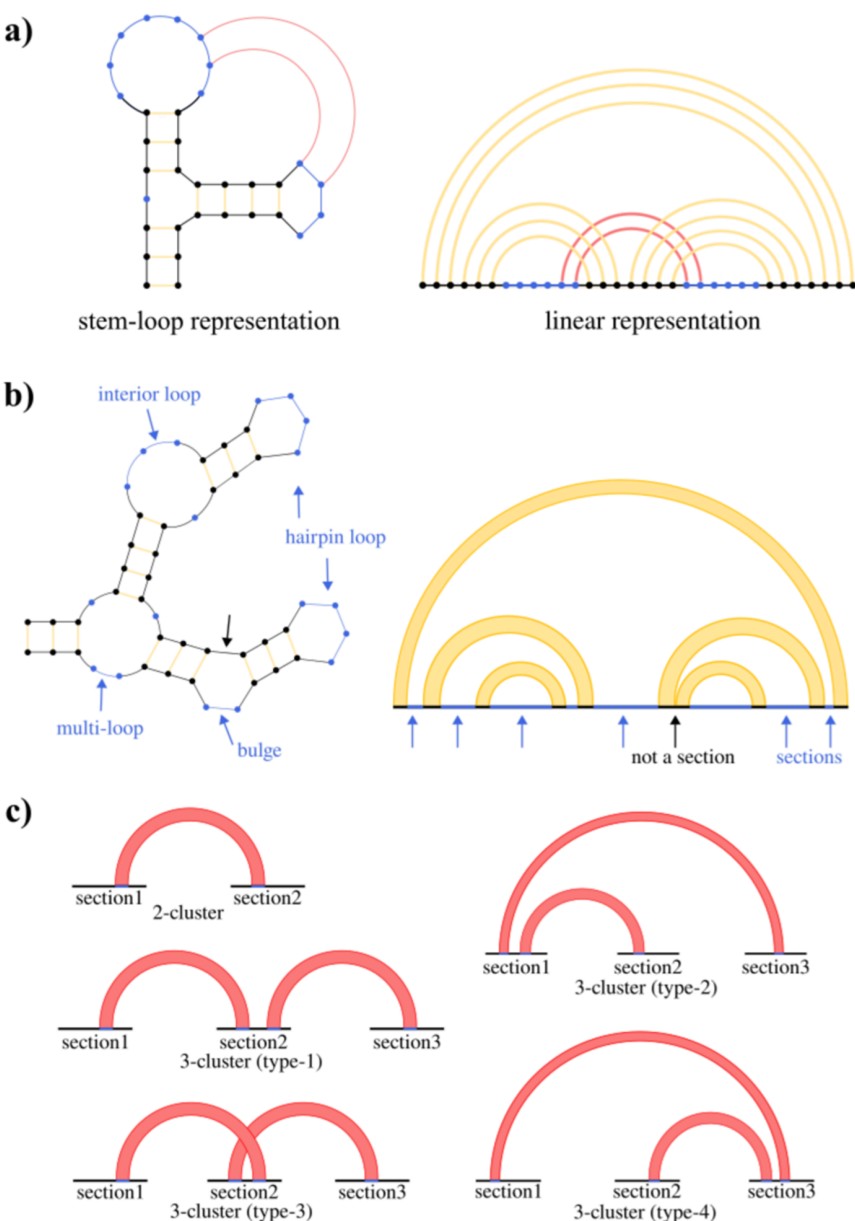

**Fig 1. Graphical representations of RNA structures and section-based pseudoknot classifications.** Yellow lines: secondary structure base pairs; red lines: pseudoknot base pairs; blue regions: sections (unpaired nucleotides); black regions: non-sections. a) RNA structure in stem-loop and linear representations. b) Stem-loop (left) and linear (right) representations of the same structure. Blue arrows indicate sections (contiguous unpaired regions). The black arrow indicates a non-section where nucleotides participate in secondary structure pairing. c) Topologically distinct pseudoknot structures of both 2-clusters and 3-clusters.

However, in this work, the contribution of loops that include base pairs in secondary structures are neglected, and only loops that include pseudoknot base pairs are considered. Hence, the free energy contribution of the psuedoknot in Fig 2 is calculated as

(FE contribution of the pseudoknot in Fig 2A))

= (stacking (4)) + (bulge loop (5)) + (stacking (6)) + (multi loop (7)) + (stacking (8)).     (3)

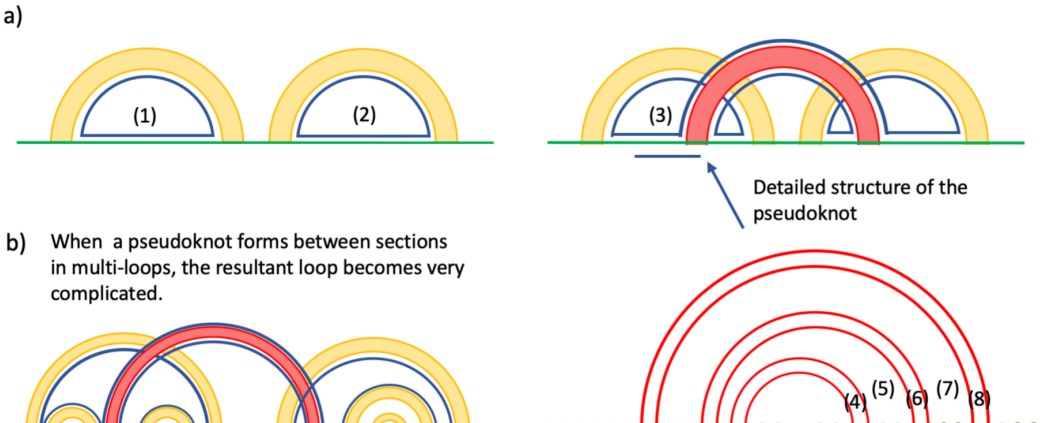

**Fig 2**. **Energy model calculation examples demonstrating local versus global structural considerations.** a) Pseudoknot base pair structure. The yellow bands represent sets of base pairs belonging to a secondary structure, while the red band represents a set of consecutive pseudoknots. Bottom right shows detailed base pair structure in the pseudoknot. b) Pseudoknot connecting sections across multi-loops. The newly formed loop that includes the pseudoknot contains all sections from the connected multi-loops, requiring global rather than local energy calculations.

Our energy model uses two main nearest-neighbour thermodynamic parameters: stacking energies and loop energies. Stacking energies represent free energy contributions of stacked pairs formed by consecutive base pairs, with values ranging from –3.4 kcal/mol (most stable) to +1.5 kcal/mol (least stable). Loop energies compose of multiple parameter arrays such as destabilising values for interior loops (4.1 to 7.4 kcal/mol for lengths 1–30), bulge loops energies (3.9 to 6.7 kcal/mol), mismatch energy parameters (–2.7 to +0.2 kcal/mol) [21]. We exclude free energy contributions from loops containing secondary structure base pairs to focus on local interactions between sections under analysis. This exclusion is essential for our hierarchical approach. For example, when sections within our target pair reside in a multi-loop (Fig 2B), calculating their local pseudoknot structure would necessitate predicting pseudoknot formations across other section pairs simultaneously, forcing expansion from local to global structure prediction and defeating the purpose of our section-based approach. Moreover, global base pair structures predicted by DPA are known to be inaccurate even with additional energy parameters for pseudoknot motifs [49], likely due to inability to account for 3D conformation effects [60, 61]. Rather than incorporating 3D conformational effects into DPA, we focus on predicting local structure using only local information.

Our energy model further simplifies entropic contributions by treating loop contributions as purely additive. This additive approximation, independent analysis of section pairs, and loop exclusion achieve $O(n^2\ell^4)$ computational cost while maintaining focus on local interactions. The empirical validity of these approximations is demonstrated in our Results section. For exact thermodynamic parameter values, refer to our code. For exact thermodynamic parameter values used in this study, refer to our Github.

While our approach for energy calculations shares mathematical similarities with RNA hybridization algorithms through nearest-neighbor dynamic programming models, there are key differences in our implementation. Our method specifically addresses intramolecular interactions between sections within a single RNA molecule, unlike hybridization algorithms that focus on intermolecular binding [62,63]. Our approach also incorporates topological constraints specific to pseudoknot formation within an existing RNA secondary structure. For 3-clusters analysis, we apply weighted energy minimization (as shown in Fig 6, where $w = 0.8$ proved optimal), a consideration not typically present in standard hybridization algorithms [64–66]. Furthermore, our method distinguishes between different pseudoknot topologies (Fig 1C) and prevents crossing pseudoknot interactions within section pairs. We apply these techniques to analyse local pseudoknot interactions within known unpaired regions, which helps demonstrate how RNA structures can be examined in a hierarchical manner.

## Algorithm

Our algorithm determines the MFE structure between section pairs by considering all possible structures where base pairs can form pseudoknots with the secondary structure, constrained so that pseudoknot base pairs within the same section pair cannot cross each other. This allows pseudoknots between different sections while preventing nested pseudoknots within the same section pair. Moreover, pairing between two sections from the same loop is prohibited because such base pairs would be nested rather than pseudoknotted. Sections are identified by analyzing the loop regions from the predicted secondary structure, where each continuous stretch of unpaired nucleotides within a loop forms a distinct section. The algorithm systematically identifies all sections by scanning the RNA sequence and grouping consecutive unpaired bases that belong to the same loop structure. Fig 3 shows the flowchart of our proposed algorithm.

We denote the two sections in the given section pair as "section1" and "section2", with length $\ell_1$ and $\ell_2$ respectively. We define a matrix $C$ whose entry $C(i,j)$ ($1 \le i \le \ell_1$, $1 \le j \le \ell_2$) represents the minimum value of free energy contribution over candidate structures in which $i$th base in section1 and $j$th base in section2 are paired and $i'$th base in section1 or $j'$th base in section2 are not connected when either $i' < i$ or $j < j'$. We set $C(i,j) = \infty$ when the $i$th base in section1 and the $j$th base in section2 cannot form a base pair. By definition, the MFE between the section-pairs that we are calculating is the minimum entry of the $C$ matrix. Thus, the problem of finding the MFE turns into the calculation of matrix $C$. The entries of matrix C can be calculated using the following recursion relation.

$$C(i,j) = \min \begin{cases} C(i+1, j-1) + (\text{stacking energy}) & \text{if } (i \ne \ell_1,\ j \ne 1) \\ C(i+d, j-1) + (\text{bulge loop energy}) & \text{if } (2 \le d \le \ell_1 - i,\ j \ne 1) \\ C(i+1, j-d) + (\text{bulge loop energy}) & \text{if } (i \ne \ell_1,\ 2 \le d \le j-1) \\ C(i+d_1, j-d_2) + (\text{interior loop energy}) & \text{if } (2 \le d_1 \le \ell_1 - i,\ 2 \le d_2 \le j-1) \\ 0 & \text{(if the base pair is the innermost one)} \end{cases} \tag{4}$$

After calculating the $C$ matrix (see Algorithm 1), we determine the MFE structure between section pairs by selecting the binding sites through dynamic programming that minimizes free energy, where the algorithm identifies the minimum entry in the $C$ matrix, representing the optimal base pairing configuration. Base pairs are formed only between complementary bases (AU, UA, CG, GC, GU, UG). This restriction to canonical pairs (Watson-Crick and wobble) is due to the availability of thermodynamic parameters in mfold 3.6. Analysis of our datasets shows canonical pairs constitute over 99% of pseudoknot pairs in transfer messenger RNA (tmRNA) and over 90% in ribonuclease P RNA (RNase P RNA). For sections to interact, they must be from different loops, contain complementary bases, and have negative free energy contribution due to their interaction. For 2-clusters, we use the MFE structure between the section pair. For 3-clusters, we apply weighted energy minimization: (energy of former pair) + $w \times$ (energy of latter pair), where $0 < w \le 1$. Sections are ordered by their position in the RNA sequence (5' to 3' direction), with the former section pair connecting sections appearing earlier and the latter section pair connecting sections appearing later. This ordering reflects co-transcriptional folding, where 5' regions can form interactions before 3' regions are fully synthesized [67,68]. We ensure consistency by preventing bases already paired in former sections from forming pairs in latter sections. This hierarchical approach constructs the global structure by first identifying favorable local interactions, then combining them while maintaining structural constraints.

We prove the computational complexity of our algorithm is $O(n^2 l^4)$. If $n$ is the number of sections and $l$ the typical length of one section, we compute the MFE for all possible section pairs for $n$ sections, yielding $n(n-1)/2 \approx O(n^2)$. For each section pair with lengths $l_1$ and $l_2$, we construct matrix $C$ of size $l_1 \times l_2$. For each entry $C(i,j)$, there are three cases: stacking with $O(1)$ complexity, bulge loop with $O(l_1)$ and $O(l_2)$ complexity respectively, interior loop with $O(l_1 \times l_2)$ complexity due to two nested loops of length $l_1$ and $l_2$. Thus, the worst-case time complexity for each $C(i,j)$ entry is $O(l_1 \times l_2 \times l_1 \times l_2)$ or $O(l^4)$

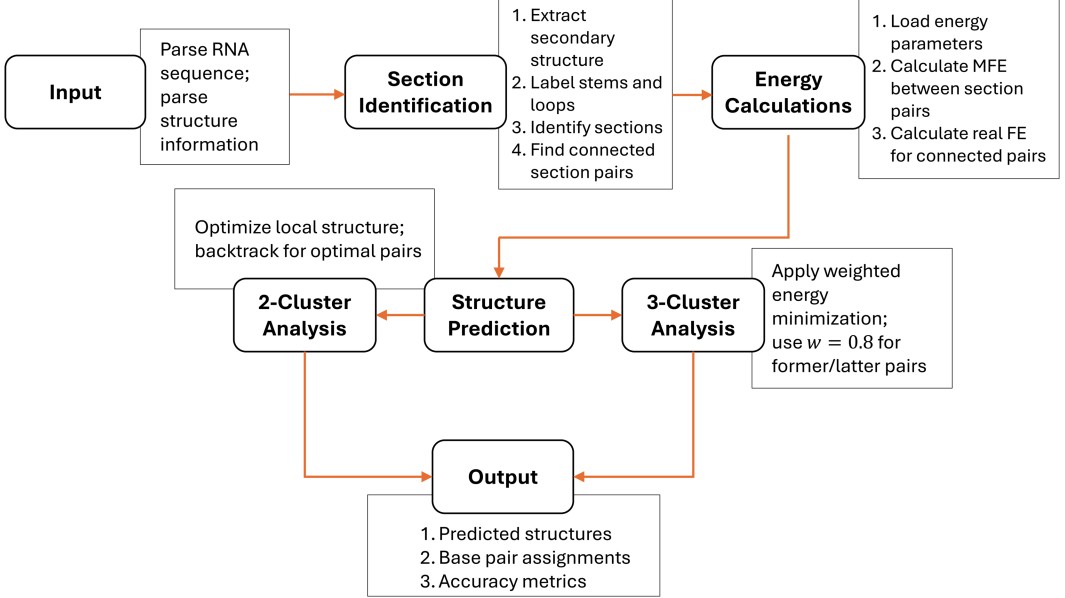

**Fig 3**. **Implementation flowchart of the RNA pseudoknot structure prediction algorithm.** The flowchart illustrates the hierarchical approach of our method, starting with input parsing, followed by section identification from the secondary structure. Energy calculations determine the minimum free energy (MFE) between section pairs. Structure prediction then branches into 2-cluster and 3-cluster analysis paths, with the latter using weighted energy minimization ($w = 0.8$) to balance former and latter section pair contributions. The output includes predicted structures, base pair assignments, and accuracy metrics.

assuming $l_1 \approx l_2 \approx l$. Thus, for all $O(n^2)$ section pairs, the total computational cost is $O(n^2 l^4)$. For typical RNA molecules in our dataset, we observed $n \approx N/l$, where $N$ is the total RNA length. This relationship suggests our algorithm's complexity in terms of total RNA length is approximately $O(N^2)$.

Our algorithm was implemented in C and compiled with gcc optimization flags. Runtime analysis on our complete dataset (1,180 sequences, 102–1,331 nt) showed a total execution time of 7.54 seconds, with mean runtime of 6.39 ms per sequence (median: 5.35 ms, IQR: 3.98–6.65 ms). Only 6.8% of sequences were classified as statistical outliers beyond 1.5×IQR, demonstrating consistent performance suitable for large-scale RNA database analysis.

## Results and discussion

In this work, we analyzed 726 tmRNA sequences and 454 RNase P RNA sequences from the RNAstrand database [69]. These RNA families were selected because they represent a comprehensive and diverse dataset of biologically relevant structures with well-documented pseudoknots. Rather than selecting a small subset of examples, we analyzed the complete set of these RNA families available in the database to ensure a thorough and unbiased evaluation of our approach. Both families consist of relatively long RNA sequences with many pseudoknots, making them ideal test cases for our section-based prediction method.

### Basic properties of investigated RNA sequences

Table 1 summarises key characteristics of tmRNA and RNase P RNA sequences from the RNAstrand database. Analysis of the RNAstrand database reveals structural similarities between the two RNA classes (Table 1). Mean sequence lengths are 368 nt (tmRNA) and 333 nt (RNase P RNA), with ~30 sections per sequence and mean section lengths of 8 nt and 5 nt respectively. tmRNA sequences show greater variability in their measurements. The increase in standard deviation is

**Algorithm 1 Dynamic programming algorithm for MFE calculation between section pairs.**

$\ell_1$, $\ell_2$: section lengths; $s_1$, $s_2$: starting positions; $C$: DP matrix; $can\_pair$: possible base pairs; $mismatches$: mismatch types between bases

```
1:  function SEGMENT_MFE(segments, loops, bases, energy_params, ℓ, seg₁, seg₂)
2:      l₁,l₂ ← get_section_lengths(segments, ℓ, seg₁, seg₂)
3:      s₁,s₂ ← get_section_starting_points(segments, ℓ, seg₁, seg₂)
4:      if loops[s₁] = loops[s₂] then return ∞
5:      end if                // Sections in same loop cannot form pseudoknots
6:      Initialize can_pair[l₁][l₂] and mismatches[l₁][l₂]
7:      for i ← 0 to l₁ − 1 do
8:          for j ← 0 to l₂ − 1 do
9:              can_pair[i][j] ← pair_type(bases[s₁ + i], bases[s₂ + j])
10:             mismatches[i][j] ← mismatch_type(bases[s₁ + i], bases[s₂ + j])
11:         end for
12:     end for
13:     Initialize C[l₁][l₂] and set MFE ← 0.0
14:     for i ← l₁ − 1 downto 0 do
15:         for j ← 0 to l₂ − 1 do
16:             if can_pair[i][j] = 0 then
17:                 C[i][j] ← ∞            // i,j cannot pair
18:             else
19:                 current_energy ← 0.0          // Case: i,j is the first bond
20:                 if i < l₁ − 1 and j > 0 then        // Stacking
21:                     current_energy ← min(current_energy, C[i + 1][j − 1] + stacking(can_pair[i + 1][j − 1], can_pair[i][j]))
22:                 end if
23:                 if i < l₁ − 1 then        // Bulge loop case 1
24:                     for d ← 2 to j do
25:                         current_energy ← min(current_energy, C[i+1][j−d]+bulge_loop(d−1,can_pair[i+1][j−d], can_pair[i][j], mismatches[i][j − d + 1], mismatches[i + 1][j − 1]))
26:                     end for
27:                 end if
28:                 if j > 0 then        // Bulge loop case 2
29:                     for d ← 2 to l₁ − i do
30:                         current_energy ← min(current_energy, C[i+d][j−1]+bulge_loop(d−1,can_pair[i+d][j−1], can_pair[i][j], mismatches[i + d − 1][j], mismatches[i + 1][j − 1]))
31:                     end for
32:                 end if
33:                 for all d₁ ∈ [2,l₁ − i], d₂ ∈ [2,j] do            // Interior loop
34:                     current_energy ← min(current_energy, C[i + d₁][j − d₂] + interior_loop(d₁ − 1, d₂ − 1, can_pair[i + d₁][j − d₂], can_pair[i][j], mismatches[i + d₁ − 1][j − d₂ + 1], mismatches[i + 1][j − 1]))
35:                 end for
36:                 C[i][j] ← current_energy
37:                 MFE ← min(MFE, current_energy)
38:             end if
39:         end for
40:     end for
41:     return MFE
42: end function
```

due to sequences exceeding 1000 bases, with each sequence containing a single extended section that forms no pseudoknot connections with other sections. These outlier sequences do not impact our analysis of pseudoknot interactions between section pairs. The substantial length of these RNA sequences presents a challenge, as conventional prediction methods struggle to accurately determine the structure of such extended sequences [36].

**Table 1. Structural characteristics of RNA sequences analyzed from RNAstrand database.** Section length refers to the number of unpaired nucleotides within each contiguous unpaired region.

| Parameter | tmRNA | RNase P RNA |
|---|---|---|
| Sequences analyzed (n) | 726 | 454 |
| Mean sequence length $\pm$ SD (nt) | 367.7 $\pm$ 86.3 | 332.6 $\pm$ 49.6 |
| Median sequence length (nt) | 363 | 330 |
| Sequence length Q1–Q3 (nt) | 353–370 | 301–365 |
| Mean sections per sequence | 29.8 | 28.1 |
| Median sections per sequence | 30 | 28 |
| Mean section length $\pm$ SD (nt) | 7.7 $\pm$ 23.1 | 5.2 $\pm$ 6.1 |
| Median section length (nt) | 4 | 4 |

## Predicting the connected section pairs based on MFE

Fig 4A shows the distribution of MFE gain (absolute value of MFE) for all section pairs. Frequency decays exponentially with increasing MFE gain. The exponential decay follows from two assumptions. Firstly, the probability of forming each additional base pair between two sections is independent of existing structure ("probability of base pairing"). Secondly, each base pair contributes approximately 1.0 kcal/mol to the free energy, representing the average stacking energy in our model. Under these assumptions, the histogram in Fig 4 fits an exponential function with parameter $a$:

$$\text{frequency} \propto \exp(-a \times (\text{MFE gain})). \tag{5}$$

The probability of base pairing can be calculated as

$$\begin{aligned}\text{probability of base pairing} &= \exp(-a \times (\text{free energy contribution per one base pair})) \\ &= \exp(-a \times 1.0\,(\text{kcal/mol})).\end{aligned} \tag{6}$$

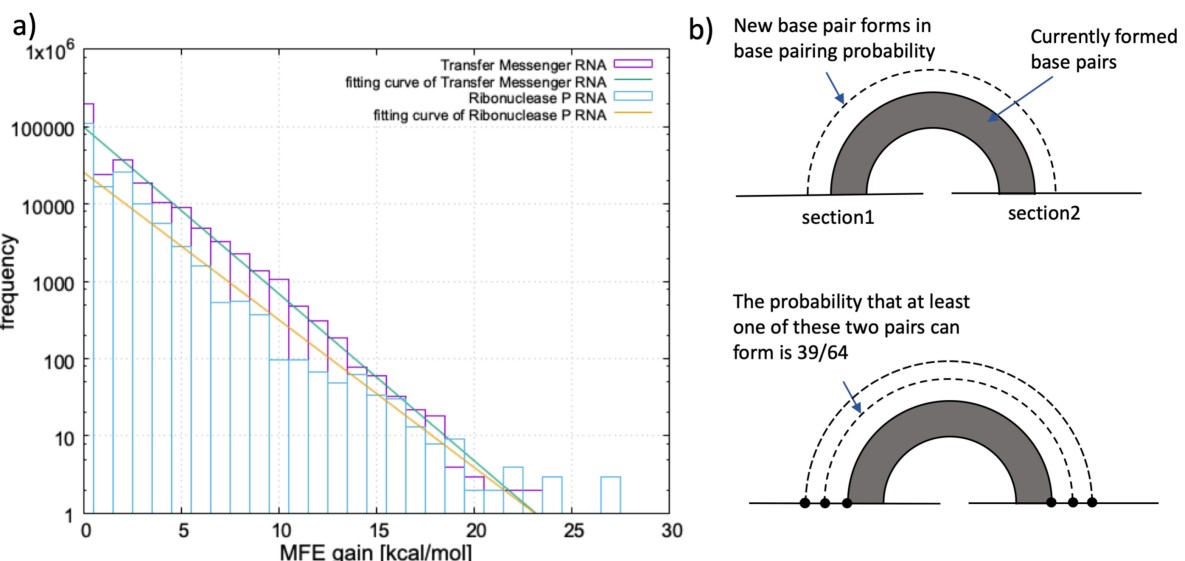

**Fig 4. Distribution of MFE gain for section pairs in RNA databases.** a) Histogram showing exponential decay of MFE gain frequency for all section pairs in tmRNA and RNase P RNA sequences from RNAstrand database. b) Simple probabilistic model explaining the observed exponential decay, assuming independent base pairing probability and ~1.0 kcal/mol energy contribution per base pair.

Curve fitting analysis reveals base pairing probabilities of 0.608 for tmRNA and 0.632 for RNase P RNA. These values reflect fundamental RNA base pairing constraints. With six possible pairing types (AU, UA, CG, GC, UG, GU), the probability of two adjacent bases forming a new base pair is 3/8 (0.375). The probability of at least one of two adjacent base pairs forming next to an existing pair is $1 - (5/8)^2 = 39/64$ (0.609), matching our observed pairing probabilities.

Our results show that MFE structures between section pairs primarily consist of stacking interactions and small internal loops of length 2. The exponential decay in the distribution reflects the rarity of section pairs with large MFE gain. Specifically, only 5.5% of pseudoknots in tmRNA and 2.8% in RNase P RNA have MFE gain exceeding 5.0 kcal/mol. Even fewer exceed 10.0 kcal/mol: just 0.47% in tmRNA and 0.23% in RNase P RNA.

We investigated the connecting probability for both tmRNA and RNase P RNA in our database (Fig 5). Connecting probability is defined as the ratio between connected section pairs and all section pairs with a given MFE:

$$\text{Connecting probability} = \frac{\text{Total number of connected section pairs with MFE}}{\text{Total number of all section pairs with MFE}}, \qquad (7)$$

Connecting probability correlates with high MFE gain as expected, since larger energy gains stabilize structures. However, we observed non-connecting section pairs with unexpectedly large MFE gains. These sections have large total MFE gain but small MFE gain per base pair compared to genuinely connected pairs. The loss of conformational entropy, which our nearest-neighbor energy model ignores, outweighs the stabilizing effect of each individual pair.

Connected section pairs concentrate among those with large MFE gains, which represent a small fraction of all section pairs. We sorted all section pairs by MFE gain and examined the cumulative distribution of connected pairs. We find that more than 90% of connected section pairs in tmRNA are included in the top $10^4$ section pairs (3%) with largest maximum free energy gains (out of all the $3.5 \times 10^5$ possible section pairs). Over 90% of connected section pairs fall within the top

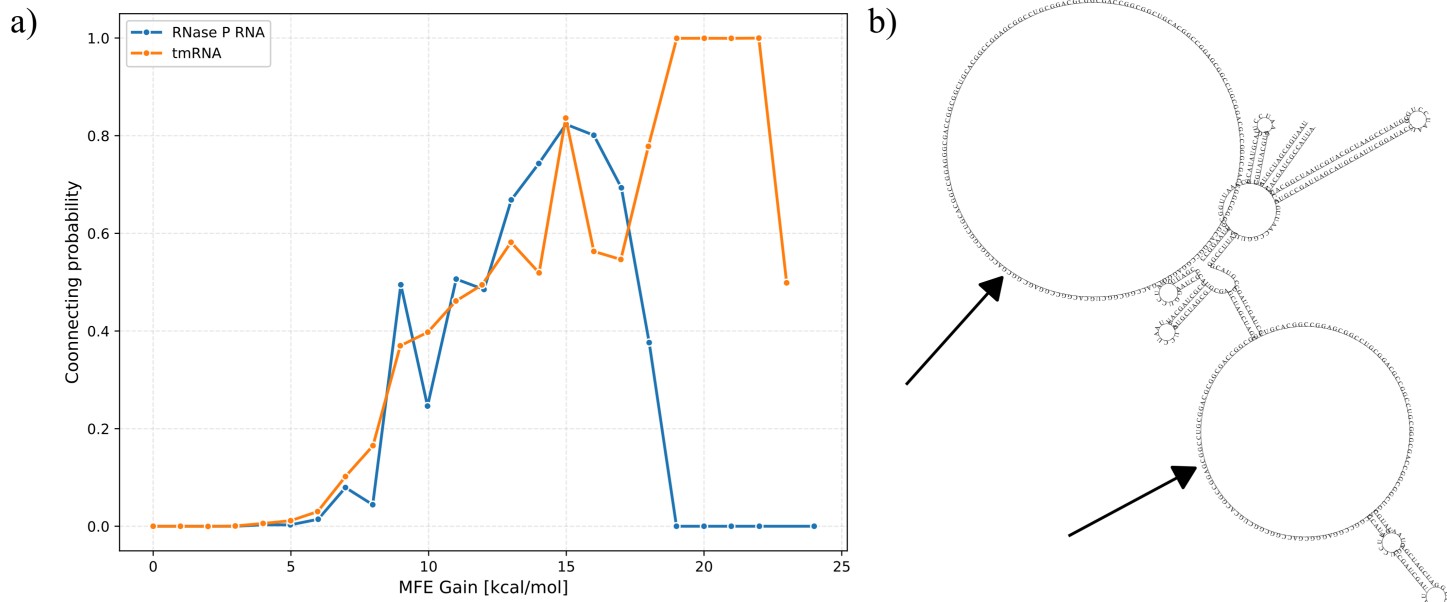

**Fig 5**. **Analysis of pseudoknot formation likelihood based on energetic predictions.** a) Connecting probability as a function of MFE gain for tmRNA and RNase P RNA sequences, defined as the ratio of actually connected section pairs to all section pairs with a given MFE value. b) Representative example of a non-connecting section pair with substantial MFE gain (80 kcal/mol), illustrating how conformational entropy losses can outweigh local energy gains in long sections ($\sim$104 and $\sim$48 nucleotides).

3% by MFE gain for tmRNA ($10^4$ of $3.5 \times 10^5$ pairs) and top 1% for RNase P RNA (1,500 of $1.8 \times 10^5$ pairs). Therefore, it is sufficient to only consider a small fraction of section pairs with large MFE gain in order to predict RNA structures with pseudoknots. On average, we only need to consider $\sim$14 section pairs per tmRNA sequence (10,000/726) or $\sim$3 section pairs per RNase P RNA sequence (1,500/454) to account for 90% of pseudoknots.

**Comparison between MFE structures and real structures for connected section pairs**

Connected section pairs exclusively form either 2-clusters or 3-clusters, with no observed instances of larger cluster formations (Table 2). Structures with pseudoknot order 1(2-clusters: 2337 in tmRNA, 543 in RNase P RNA) substantially outnumber structures with pseudoknot order 2 (3-clusters: 110 in tmRNA, 1 in RNase P RNA). This distribution is consistent with hierarchical folding, where lower-order pseudoknots form before higher-order structures. We compared MFE predictions with actual free energies (real FE) for connected 2-cluster section pairs. While the proportion of 2-clusters where real FE = MFE varies considerably between RNA classes (68.7% for tmRNA versus 39.9% for RNase P RNA), >80% of all 2-clusters demonstrate free energy gain >0.8 × MFE gain.

We compared MFE-predicted base pair arrangements with actual structures for 2-clusters. For each cluster, we calculated sensitivity (TPR) and positive predictive value (PPV) using true positives (TP), false positives (FP), and false negatives (FN). When multiple MFE structures existed, we averaged these metrics:

$$\text{TPR(sensitivity)} = \frac{TP}{TP + FN}, \quad \text{PPV} = \frac{TP}{TP + FP}. \tag{8}$$

With sensitivity >0.9 and PPV >0.8 for both RNA classes, our algorithm outperforms global predictions methods, which typically achieves a sensitivity score of 0.5–0.7 and PPV <0.7 for complex pseudoknotted structures [45].

It is important to acknowledge that our analysis relies on knowledge of the true secondary structure from which we derive the unpaired sections. This differs from the more challenging problem of *de novo* pseudoknot prediction without prior structural information. While our results demonstrate that local energetic considerations can effectively predict pseudoknot formations between known unpaired regions, additional research would be needed to determine whether similar accuracy could be achieved when starting with predicted rather than true secondary structures.

3-clusters exhibit several topologically distinct configurations (Fig 1C), with 110 type-1 3-clusters in tmRNA and only one type-2 3-cluster in RNase P RNA. Given limited type-2 samples, we analyze only type-1 3-clusters, designating section1-section2 (see Fig 1C) connections as "former pairs" and section2-section3 as "latter pairs".

We analysed the 110 type-1 3-clusters in tmRNA by comparing MFE calculations with actual structures for both pair types, treating them independently (similar to 2-cluster analysis). For former section pairs, approximately 50% have real structures identical to predicted MFE structures, while over 80% have free energy contributions lower than 0.8×MFE (Table 3). These results closely resemble our 2-cluster prediction accuracy. In contrast, latter section pairs show markedly different behavior, with only a small fraction having free energy contributions approximating their MFE. Former section pairs demonstrate prediction accuracy comparable to 2-clusters, while latter section pairs exhibit particularly low PPV

**Table 2**. The result of analyses based on section pairs.

| Parameter | tmRNA | RNase P RNA |
|---|---|---|
| The number of all 2-clusters | 2,337 | 534 |
| The number of all 3-clusters | 110 | 1 |
| The number of 2-clusters for which MFE = real FE | 1,605 (68.7%) | 213 (39.9%) |
| The number of 2-clusters for which real FE < 0.8×MFE | 1,972 (84.4%) | 438 (82.0%) |
| Sensitivity of base pair prediction for 2-clusters | 0.9159 | 0.9087 |
| PPV of base pair prediction for 2-clusters | 0.8400 | 0.8056 |

**Table 3. The comparison of MFE and free energy contribution of real structure for connected section pairs in 3-clusters.**

| Parameter | Former section pair | Latter section pair |
|---|---|---|
| Total number | 110 | 110 |
| The number of section pair for which real FE = MFE | 55 (50.0%) | 10 (9.1%) |
| The number of section pair for which real FE<0.8×MFE | 91 (82.7%) | 18 (16.3%) |
| Sensitivity of base pair prediction | 0.8343 | 0.5529 |
| PPV of base pair prediction | 0.7164 | 0.2568 |

values. Across both pair types, sensitivity consistently exceeds PPV, indicating our method tends to predict excess base pairs. This over-prediction is expected, as our approach treats these connections as independent 2-clusters despite their presence in more complex 3-cluster structures.

The failure suggests that former section pairs form independently, whereas latter section pair formations are strongly influenced by pre-existing base-pair connections. This asymmetry appears to reflect the fundamental biology of RNA synthesis, which proceeds directionally from 5' to 3' along the sequence. During transcription, former section pairs likely establish their interactions before latter sections are even fully synthesised, as supported by previous research on co-transcriptional folding dynamics [70,71]. The consistency of this pattern across our dataset suggests it represents a general feature of RNA folding rather than a family-specific characteristic. These findings challenge the assumption that RNA structures in cells form solely through global free energy minimization [72–75]. Instead, our results suggest that the dynamic and sequential nature of RNA synthesis/folding plays a crucial role in determining the final structural configuration, particularly for complex pseudoknot arrangements. This sequential dependency within pseudoknot order 2 structures demonstrates that even within a single higher-order pseudoknot, simpler interactions must establish before complex interactions can form.

To evaluate whether our method's accuracy depends on sequence length, we analyzed all RNA sequences from our dataset. For each sequence, we compared predicted connections (negative MFE values) against actual connections in known structures. We calculated the true positive rate (TPR) and positive predictive value (PPV) for each sequence and examined their correlation with sequence length using Pearson correlation analysis.

For TPR versus sequence length, $r = -0.026$ ($p = 0.392$), while for PPV versus sequence length, $r = 0.046$ ($p = 0.129$). Both metrics show negligible correlation with sequence length ($|r| < 0.05$) with high $p$-values ($p > 0.05$), confirming no significant dependency. Under the null hypothesis of no correlation, these results support the conclusion that prediction accuracy remains consistent across the tested sequence length range.

The prediction accuracy independence is due to our hierarchical approach. Our algorithm calculates minimum free energy locally between section pairs rather than globally across the entire RNA structure. Consequently, prediction accuracy is determined by local section-pair interactions and section length $\ell$ rather than total RNA length $L$, consistent with our $O(n^2\ell^4)$ computational complexity. For a given section length, computational cost scales with the number of section pairs, not with total sequence length, ensuring that prediction accuracy remains stable regardless of overall RNA size.

### The comparison between MFE structures and real structures for 3-clusters

Our previous analysis treated section pairs independently, ignoring that bases in section2 cannot simultaneously pair with both section1 and section3. We now predict complete 3-cluster structures while enforcing this section pair dependency with two prediction strategies:

1. Equal weighting: Minimize the combined free energy contribution from both former and latter section pairs, treating them with equal importance.
2. Sequential determination: First identify the minimum free energy structure of the former section pair, then determine the latter section pair structure while ensuring no bases in section2 are reused.

These approaches represent special cases of a weighted method that minimizes:

$$\text{(free energy contribution of former pair)} + w \times \text{(free energy contribution of latter pair)}, \tag{9}$$

where $0 < w < 1$. We implemented an energy minimisation algorithm that evaluates all possible base pairings between sections (Fig 6), calculating MFE contributions for each former-latter pair combination using nearest-neighbor thermo-dynamic parameters from mfold 3.6. The algorithm considers stacking interactions, bulge loops, and interior loops while preventing bases already paired in the former pair from participating in latter pair interactions. We determined optimal $w$ by testing values from 0 to 1, calculating weighted energies, predicted base pairs, and accuracy metrics (sensitivity and PPV) for each. Optimal performance occurs at $w = 0.8$, indicating preferential weighting of former pairs while maintaining latter pair contributions. However, 3-cluster latter pair prediction remains poor across all $w$ values, suggesting their formation is influenced by global structure or 3D conformation beyond our local energy model's scope.

A limitation of our approach is the use of energy parameters derived from salt-free experimental conditions (mfold 3.6) to predict structures that typically form in the presence of salt. This approximation was necessary due to the lack of comprehensive energy tables for RNA in various salt concent rations. Despite this limitation, our method demonstrates strong predictive power for 2-clusters, suggesting that local pseudoknot formation may be reasonably approximated by these parameters even without explicitly modeling ionic effects. Future work could significantly improve prediction accuracy by incorporating salt-dependent energy parameters as they become available.

Our method is successful despite using additive entropy approximation for pseudoknots. Although previous work has shown that entropy contributions in pseudoknotted structures are not strictly additive [76,77], our results for 2-clusters (sensitivity >0.90, PPV >0.80) suggest this simplification works well within our hierarchical section-based framework.

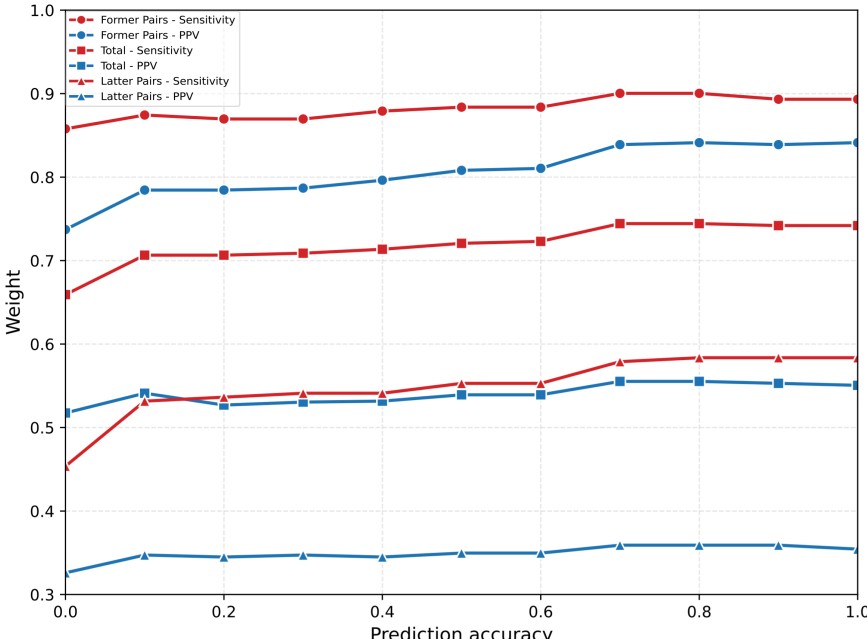

**Fig 6. Optimization of weighting parameter for 3-cluster pseudoknot structure prediction.** Prediction accuracy metrics (sensitivity and positive predictive value) as a function of weight parameter $w$ applied to the free energy contribution of latter section pairs in the weighted energy minimization approach. Optimal performance occurs at $w = 0.8$, indicating preferential weighting of former section pair interactions while maintaining contribution from latter pairs in type-1 3-cluster configurations.

The high accuracy indicates that for common pseudoknot configurations, local energy minimization with simplified entropy terms captures the essential physics governing structure formation. The method's limitations with latter section pairs in 3-clusters likely indicate where more complex entropic considerations become necessary. Nonetheless, this simplification allows for structural prediction of longer RNA sequences while maintaining good accuracy for the most common pseudo-knot configurations.

## Conclusion

We have presented a section-based approach to RNA pseudoknot analysis that employs mfold nearest-neighbor energy model with DPA. This method offers both computational efficiency and prediction accuracy advantages over traditional approaches. Our algorithm scales as $O(n^2\ell^4)$, where $n$ represents the number of sections and $\ell$ represents the average section length. Assuming section length remains independent of total RNA sequence length, the computational complexity reduces to $O(L^2)$ for an RNA of length $L$. A key finding of our research is the highly concentrated distribution of biologically relevant pseudoknots among section pairs with large MFE gains. By focusing on just the top 3% of section pairs ranked by MFE gain, we can capture approximately 90% of all connected section pairs. This observation enables substantial reductions in the search space required for structure prediction.

Our approach predicts pseudoknot order 1 structures (2-clusters) with sensitivity above 0.90 and positive predictive value exceeding 0.80. These results significantly outperform conventional methods that attempt to predict global pseu-doknotted structures in a single step. This indicates that once unpaired regions are identified, 2-cluster pseudoknot for-mations between them appear to be primarily determined by local information. This observation suggests potential value in hierarchical approaches to RNA structure prediction, though such approaches would need to address the challenge of accurately identifying relevant unpaired regions and resolving potential competition between pseudoknot and regular secondary structure formation. Our work provides evidence for the effectiveness of local energy calculations specifically for the subproblem of pseudoknot prediction between known unpaired regions. However, significant limitations exist for pseudoknot order 2 structures (3-clusters). Nearly all 3-clusters conform to the type-1 configuration (Fig 1C). While our approach can predict former section pairs in these 3-clusters with accuracy comparable to 2-clusters, it performs poorly when predicting latter section pairs. The inadequacy of our local energy model for latter section pairs suggests that these pseudoknots are significantly influenced by factors beyond local interactions, such as the dynamic process of RNA fold-ing and the global 3D RNA conformation. These findings highlight the limitations of purely thermodynamic approaches to complex RNA structure prediction and suggest that accurate modeling of 3-clusters may require incorporating kinetic folding pathways and long-range structural constraints.

## Acknowledgments

R.M., D.L., and E.H.Y thank Dr. Michael Zuker for his permission to use the mfold energy model that he developed.

## Author contributions

**Conceptualization:** Ryota Masuki, Ee Hou Yong.

**Data curation:** Ryota Masuki.

**Formal analysis:** Ryota Masuki, Ee Hou Yong.

**Funding acquisition:** Ee Hou Yong.

**Investigation:** Ryota Masuki, Ee Hou Yong.

**Methodology:** Ryota Masuki, Ee Hou Yong.

**Project administration:** Ee Hou Yong.

**Supervision:** Ee Hou Yong.

**Validation:** Ee Hou Yong.

**Visualization:** Ryota Masuki, Donn Liew, Ee Hou Yong.

**Writing – original draft:** Ryota Masuki, Ee Hou Yong.

**Writing – review & editing:** Donn Liew, Ee Hou Yong.

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
