## [Decision Letter · Decision Letter 0]

16 Oct 2025

PCOMPBIOL-D-25-01796

Hierarchical Analysis of RNA Secondary Structures with Pseudoknots Based on Sections

PLOS Computational Biology

Dear Dr. Yong,

Thank you for submitting your manuscript to PLOS Computational Biology. After careful consideration, we feel that it has merit but does not fully meet PLOS Computational Biology's publication criteria as it currently stands. Therefore, we invite you to submit a revised version of the manuscript that addresses the points raised during the review process.

Please submit your revised manuscript within 60 days Dec 16 2025 11:59PM. If you will need more time than this to complete your revisions, please reply to this message or contact the journal office at ploscompbiol@plos.org. Please include the following items when submitting your revised manuscript:

We look forward to receiving your revised manuscript.

Kind regards,

Arli Aditya Parikesit, PhD

Academic Editor

PLOS Computational Biology

Arne Elofsson

Section Editor

PLOS Computational Biology

**Additional Editor Comments:**

Based on reviewers’ reports, it is clear that there should be extensive revision for the manuscript. These are pointers from the reports, for the revision:

1. Consider different scenario for the optimization of the algorithm.

2. Report the computational cost of the pipeline

3. Deposition of the source code and data

4. Authors are not required to cite proposed articles by the reviewer if they are irrelevant with the manuscript

5. Strengthen the benchmarking foundation of the pipeline

6. Use standard terminology in RNA structure research

7. Make sure that the tables and figures are coherent, plausible, and align well to the main text.

8. Consider different scenarios for the structure optimization.

**Journal Requirements:**

4) Please ensure that the funders and grant numbers match between the Financial Disclosure field and the Funding Information tab in your submission form. Note that the funders must be provided in the same order in both places as well.

State what role the funders took in the study. If the funders had no role in your study, please state: "The funders had no role in study design, data collection and analysis, decision to publish, or preparation of the manuscript.".

**Reviewers' comments:**

Reviewer's Responses to Questions

**Comments to the Authors:**

Reviewer #1: The manuscript presents a section-based framework for RNA pseudoknot prediction. The study is clearly structured, and the proposed approach reduces computational complexity compared with conventional methods. Overall, the manuscript makes a valuable contribution to the field of RNA structure prediction. To further strengthen the quality and clarity of the work, I suggest that the authors address the following issues:

Issues

1. The manuscript does consider 3-cluster pseudoknots, but the authors should also provide an analysis of the worst-case scenario. While the derivation of the O(n^2L^4) complexity is convincing, the discussion of 3-cluster cases could be expanded with a clearer comparison between the theoretical worst-case complexity (possibly O(n^3L^4) and the practical complexity after MFE-based screening.

2.

For 3-cluster structures, the method applies different weights to the former and latter pairs

(FE_former+wFE_latter). Since this algorithm is order-dependent, the authors should clarify how the order of section pairing is determined and justify the choice.

3.

The authors should report the actual runtime and computational cost when applying the method to the dataset of 726 tmRNA and 455 RNase P RNA sequences. This would provide a clearer picture of the practical efficiency beyond theoretical complexity.

4.

Table 1 currently summarizes sequence characteristics. It would be helpful to also provide statistics related to pseudoknot occurrence (e.g., number of pseudoknots per sequence, proportion of 2-clusters vs. 3-clusters), which are directly relevant to the study.

5. The Data and Code Availability section should include access to the test datasets used in the study.

Reviewer #2: This manuscript presents a clear, well-validated hierarchical framework for pseudoknot prediction with substantial computational gains (O(n²ℓ⁴)) and strong accuracy across large tRNA-m and RNase P datasets. The results are novel, well supported, and of broad interest to RNA structure prediction. I recommend acceptance.

Reviewer #3: Masuki et al. present a hierarchical method for predicting RNA structures with pseudoknots, a task often hindered by computational complexity. Their approach divides RNA into unpaired sections and predicts pseudoknot interactions between them using a nearest-neigbor energy model and dynamic programming. The method reduces computational cost while maintaining high accuracy, achieving over 0.90 sensitivity and 0.80 precision for the simplest (2-cluster) RNAs from the benchmark set. The manuscript is well-structured, the writing is coherent, clear, and mostly sufficiently detailed, although not in all parts of the text.

Unfortunately, the data and software presented are not available. To ensure reproducibility and transparency, both the software implementing the proposed algorithm and the datasets used in the experiments - as well as the resulting outputs- should be made publicly available through a recognized repository such as GitHub or Zenodo. Including only code snippets or pseudocode within the manuscript is inadequate and impractical for other researchers wishing to test or build upon the presented method. If the authors choose not to release the code and data, I would strongly recommend that the manuscript be rejected, as open availability is essential for verifying the results and advancing the field.

Other remarks are given below.

Major remarks:

(1) "RNA secondary structure is defined as the set of base pairs that form a planar graph, excluding pseudoknots which are nonplanar structural elements by definition" - this statement oversimplifies the definition of RNA secondary structure. In fact, multiple definitions exist, and a more general one describes secondary structure as simply specifying which bases are paired and which are unpaired, regardless of whether the pairings are nested or form pseudoknots (see Antczak et al., Bioinformatics 2018; doi: 10.1093/bioinformatics/btx783 - you will there a concept of hierarchical analysis and pseudoknot order that reflect the hierarchy of folding, which might be helpful in hierarchical examination MAsuki et l. use in their work). The definition adopted by the authors is a practical simplification, motivated by the difficulties associated with predicting pseudoknots and representing them in some - but not all - 2D structural notations (e.g. dot-bracket). This challenge has led to the development of methods for removing pseudoknots from 2D structures to facilitate autoated analysis (see Smit et al., RNA 2008; doi: 10.1261/rna.881308), but there remains no universal criterion for distinguishing "core" base pairs from those involved in pseudoknots (see Zok et al., AMCS 2019; doi: 10.34768/amcs-2020-0024). I suggest to clarify this conceptual nuance and reference these relevant works.

(2) Introduction: in the discussion of limitations and current challenges of pseudoknot prediction, I recommend referring to the work by Justyna et al. (2023, Briefings in Bioinformatics, doi: 10.1093/bib/bbad153), in which the authors benchmarked machine learning methods on experimental data, highlighting significant difficulties in accurately predicting pseudoknots compared to the prediction of canonical and non-canonical base pairs.

(3) The manuscript would benefit from a clearer discussion of whether pseudoknots in the analyzed RNA structures are formed primarily by canonical base pairs or if non-canonical interactions also play a role. This distinction is important, as numerous studies have demonstrated that non-canonical base pairs frequently occur within pseudoknot regions and significantly affect RNA folding, stability, and tertiary interactions. Clarifying which types of base pairings are considered in the hierarchical model would deepen the reader’s understanding of the method’s applicability and limitations.

(4) The use of the term "short-range" is misleading and inconsistent with standard RNA structural terminology. In RNA structure research, short-range interactions� are conventionally understood to mean base-pair interactions between nucleotides that are close in the primary sequence, not those that are topologically non-crossing. I strongly suggest to adopt the established term "nested interactions" to describe non-crossing base-pair arrangements, as it is both precise and widely used in the RNA structure-related literature. Reframing the terminology would avoid confusion and better align the manuscript with accepted definitions used across computational and structural RNA studies.

(5) "Common secondary structure motif includes hairpin loops, hairpin stems, helical duplexes, bulges, internal loops, multi-loops, and junctions" - What is the difference between multi-loop and junction? In the literature these terms mean exactly the same thing. Moreover, multi-loops/junctions are a subclass of internal loops. I suggest defining what exactly the authors mean by these terms.

(6) Figure 1b:

- The red arrow in arc diagram obscures the fragment it points to; it needs to be positioned differently.

- I suggest coloring sections and non-section in both representations

- In the stem-loop representation sections are represented by blue circles - does it mean that a section is not just a single stranded fragment but also base pairs that close a loop? If so, this is not consistent with the previously given definition of a section.

- "Linear" -> "linear"

- The authors provide all 3-cluster models, I suggest adding a representation of a 2-cluster pseudoknot as well to complete the view of what they analyse in the paper. I also suggest to separate this figure from panels (a)-(b). These should be two separate figures: one with panels (a)-(b), the other with panel (c) and 2-cluster added.

(7) The complexity of a pseudoknot can be reflected by a pseudoknot order. I suggest that the authors get familiar with this concept (see my comment number 1).

(8) "Moreover, two sections from the same loop is prohibited because the base pairs between these sections will be in secondary structures, rather than pseudoknots." - unclear sentence, reformulate… sections will be in nested structure rather than pseudoknotted…

(9) Table 1:

- explain what do you mean by section length (see also my comment numer 6).

- provide the information about the range of sequence length for both datasets; mean length is not enough in my opinion; besides - median is more informatived than mean

(10) I suppose that the accuracy of prediction is independent on the RNA sequence length. However, it should be clearly stated in the paper (or maybe I missed it?), as this is - if I am right - quite an important feature of the method.

Minor remarks:

(1) "involved in secondary structure base pairing" -> involved in base pairing

(2) Standardise the spelling: j-th or jth (but not both versions)

(3) j < j' We set C(i, j) -> j < j'. We set C(i, j)

(4) Sensitivity i also called the true positive rate (TPR). I suggest using "TPR" in formula (8)

(5) Table 2, Table 3 - insert spaces before brackets; 1605->1,605 etc

**Have the authors made all data and (if applicable) computational code underlying the findings in their manuscript fully available?**

Reviewer #1: **No:** The authors have only described the algorithm, but have not provided the test datasets or the source code.

Reviewer #2: Yes

Reviewer #3: **No:** My main comment to this work concerns data and software availability. To ensure reproducibility and transparency, both the software implementing the proposed algorithm and the datasets used in the experiments - as well as the resulting outputs - should be made publicly available through a recognized repository such as GitHub or Zenodo. Including only code snippets or pseudocode within the manuscript is inadequate and impractical for other researchers wishing to test or build upon the presented method. If the authors choose not to release the code and data, I would strongly recommend that the manuscript be rejected, as open availability is essential for verifying the results and advancing the field.

PLOS authors have the option to publish the peer review history of their article (what does this mean?). If published, this will include your full peer review and any attached files.

Reviewer #1: No

Reviewer #2: No

Reviewer #3: No

**Figure resubmission:**
---

## [Decision Letter · Decision Letter 1]

9 Jan 2026

Dear Dr Yong,

We are pleased to inform you that your manuscript 'Hierarchical Analysis of RNA Secondary Structures with Pseudoknots Based on Sections' has been provisionally accepted for publication in PLOS Computational Biology.

Best regards,

Arli Aditya Parikesit, PhD

Academic Editor

PLOS Computational Biology

Arne Elofsson

Section Editor

PLOS Computational Biology

THe authors have addressed the reviewers' concern accordingly. Therefore, I decide to accept this manuscript for publication. For the galley proof preparation, please kindly make sure that the reference style and citations of the manuscript is consistent in accordance to the journal guidelines.

Reviewer's Responses to Questions

**Comments to the Authors:**

Reviewer #1: My concerns have been addressed, and I have no further comments.

Reviewer #3: I am fully satisfied with the authors' comprehensive responses to my questions and comments, as well as with the changes made to the manuscript, including additions and corrections to the text and figures. The authors have also provided the algorithm along with the data sets. I have only one comment regarding the references: I suggest that the authors pay attention to the spelling of journal names in the reference list, as the list is formatted very poorly and chaotically: many journal names are written with lowercase letters, some names appear in multiple versions (e.g., Nucleic Acids Res, Nucleic acids research...). - official abbreviations of journal names should be used and applied consistently.

**Have the authors made all data and (if applicable) computational code underlying the findings in their manuscript fully available?**

Reviewer #1: Yes

Reviewer #3: Yes

PLOS authors have the option to publish the peer review history of their article (what does this mean?). If published, this will include your full peer review and any attached files.

Reviewer #1: No

Reviewer #3: No

---

## [Editor Report · Acceptance letter]

PCOMPBIOL-D-25-01796R1

Hierarchical Analysis of RNA Secondary Structures with Pseudoknots Based on Sections

Dear Dr Yong,

I am pleased to inform you that your manuscript has been formally accepted for publication in PLOS Computational Biology. Your manuscript is now with our production department and you will be notified of the publication date in due course.

With kind regards,

Lilla Horvath
